# Nanoremediation and Antioxidant Potential of Biogenic Silver Nanoparticles Synthesized Using Leucena’s Leaves, Stem, and Fruits

**DOI:** 10.3390/ijms25073993

**Published:** 2024-04-03

**Authors:** Christopher Santos Silva, Fernanda Maria Policarpo Tonelli, Vinicius Marx Silva Delgado, Vitória de Oliveira Lourenço, Geicielly da Costa Pinto, Lucas Santos Azevedo, Luciana Alves Rodrigues dos Santos Lima, Clascídia Aparecida Furtado, Danilo Roberto Carvalho Ferreira, Flávia Cristina Policarpo Tonelli, Adriano Guimarães Parreira

**Affiliations:** 1Biotechnological Processes Laboratory, Centro-Oeste Campus, Federal University of São João del-Rei, Divinópolis 35501-296, MG, Brazil; christopher.silva.s@outlook.com (C.S.S.); viniciusmarx2003@gmail.com (V.M.S.D.); vitorialourenco.bqi@gmail.com (V.d.O.L.); geicicosta1212@gmail.com (G.d.C.P.); flacristinaptonelli@gmail.com (F.C.P.T.); 2Phytochemistry Laboratory, Centro-Oeste Campus, Federal University of São João del-Rei, Divinópolis 35501-296, MG, Brazil; azevedolucas07@gmail.com (L.S.A.); luarsantos@ufsj.edu.br (L.A.R.d.S.L.); 3Carbon Nanostructure Chemistry Laboratory, Nuclear Technology Development Center (CDTN), Belo Horizonte 31270-901, MG, Brazil; clascidia@gmail.com (C.A.F.); danilorcferreira@gmail.com (D.R.C.F.); 4Protein Chemistry Laboratory, Centro-Oeste Campus, Federal University of São João del-Rei, Divinópolis 35501-296, MG, Brazil; aguiparreira@ufsj.edu.br

**Keywords:** green silver nanoparticles, *Leucaena leucocephala*, nanoremediation, antioxidant, green synthesis, nanotechnology, dyes, methylene blue, tartrazine

## Abstract

Synthetic dyes are persistent organic environmental pollutants that can cause extensive damage to living beings and to the ecosystem as a whole. Cost-effective, sustainable, and efficient strategies to deal with this type of pollution are necessary as it commonly resists conventional water treatment methods. Silver nanoparticles (AgNPs) synthesized using the aqueous extract from the leaves, stem, and fruits of *Leucaena leucocephala* (Leucena) were produced and characterized through UV–vis, TEM, EDS, SDL, XPS, XRD, and zeta potential, and they proved to be able to promote adsorption to remediate methylene blue and tartrazine pollution in water. The nanoremediation was performed and did not require direct exposure to sunlight or any special lamp or a specific reduction agent. The AgNPs produced using the extract from the leaves exhibited the best performance in nanoremediation and also presented antioxidant activity that surpassed the one from butylated hydroxytoluene (BHT). Consequently, it is an interesting nanotool to use in dye nanoremediation and/or as an antioxidant nanostructure.

## 1. Introduction

Environmental pollution is a global problem that threatens the survival of living beings on Earth. Environmental contaminants can be of various chemical natures; there are organic molecules (for example, pesticides and dyes), inorganic structures (such as heavy metals), and biological pollutants (such as pathogenic bacteria). These contaminants can severely and negatively impact the health of living beings and even cause their death [1,2].

The wastewater from food and textile industries, for example, commonly presents mixed pollution (containing various different pollutants), and among the organic contaminants, the dyes are almost always present. It is estimated that the textile industry applies approximately 10 thousand tons of dyes annually worldwide, and the annual production of these persistent pollutants is around 70 million tons [3,4,5,6,7].

When discarded directly into nature, without adequate pre-treatment, these industrial effluents containing dyes can cause damage to the environment and to living beings, either directly or indirectly. Dyes can cause an increase in chemical and biochemical oxygen demands; harm the photosynthetic activity of photosynthesizing organisms and impair their development; promote bioaccumulation throughout the food chain; lead to the death of fish species as a consequence of water eutrophication; and threaten human health due to the fact that they can act as mutagenic and carcinogenic substances. As persistent organic pollutants, they can last for long time periods in polluted areas, prolonging the threat and the damage and commonly resisting conventional water treatment methods [8,9,10]. Conventional treatment strategies are usually ineffective in totally removing them; to promote this removal, high-cost and more complex strategies to decolorize contaminated effluents are required [11,12].

Therefore, strategies for remediating water contaminated with synthetic dyes that are low-cost and environmentally sustainable are urgent and desirable. Low-cost adsorbents to remove dyes from industrial effluent are examples of materials that have exhibited effectiveness in this sense. They can be obtained from different types of waste materials or can be synthesized using low-cost bio-based raw materials [13].

In this sense, nanotechnology is a field of expertise that has the potential to offer efficient solutions. A diverse range of nanomaterials has already proven to be useful in environmental pollution remediation, degrading contaminants through catalysis [14], or removing them through adsorption [15].

Among these materials, it is possible to highlight the nanoparticles (NPs) [16,17,18]. Cryogels with immobilized NPs are an example of an innovative solution for water purification; they remove not only inorganic pollutants such as heavy metals, but also organic ones such as dyes. These nanocomposites exhibit not only adsorption, but also catalytic mechanisms of action [19].

Among NPs, the silver ones (AgNPs) have been reported to present interesting activities, such as antibacterial; they also show unique catalytic and electronic features, which are useful in the promotion of the nanoremediation of environmental pollution in aqueous samples, for example [20,21]. It has already been proven that these nanostructures are able to deal efficiently with dyes as environmental pollutants through adsorption [22] or catalysis [23].

AgNPs can be synthesized using different protocols: chemical, physical, physicochemical, and biological; however, when aiming for sustainability, green synthesis (also known as biogenic or biological synthesis) is recommended. Green synthesis protocols aim to have the lowest negative impact on the ecosystem possible, avoiding the use of harmful chemicals by applying renewable materials and consequently preventing the production of toxic waste. They also direct attention towards the optimization of energy use. For example, microorganisms and, more commonly, plant material (extracts and/or biomass) can be applied in protocols to generate AgNPs [18,24].

When it comes to plant extracts, phenolic acids are recognized as being, in the protocols aiming to produce metallic nanoparticles, substances that are largely responsible for the reduction of metal ions (from the ionic form commonly provided through a salt to the zero valent one) and that also act as capping agents (coating the particles and influencing their degree of dispersion, for example) [25,26,27,28].

When choosing the bio-based material to be used in a green synthesis protocol, however, it is interesting to dedicate attention to weedy, invasive species. The use of these plants can, in fact, offer a solution for the agricultural sector as they represent a problem during crop cultivation [29,30,31]. *Leucaena leucocephala* (Lam.) de Wit (Figure 1), also known as Leucena, is a species with a proven potential to be highly invasive in Brazil and other countries around the world. Native to Mexico, it was taken to other countries to be used in reforestation (to avoid erosion), and its seeds, due to the fact that they contain large amounts of proteins, can be eaten by humans [32]. The genus of this species belongs to the Leguminosae family, and in the field, it can impair the development of crops and threaten the native flora. After 2 years of Leucena’s introduction in Baopoling hill (Sanya City, Hainan Island, China), other vegetal species could not survive around it due to the highly invasive potential of *L. leucocephala* [33]. Consequently, it is important to propose useful applications of this species as effective ways to control its expansion are of special interest to those dealing with the threat it poses to native biodiversity [32,33].

Using this species, some silver nanoparticles have already been synthesized for use in different applications. Polysaccharides obtained from the seeds of Leucena, for example, allowed the production of AgNPs which are capable of contributing to milk preservation; the nanomaterial exhibited antifungal and anticancer activities [34]. AgNPs produced using Leucena’s extracts as green raw material also proved to be useful for applications in sensors to detect inorganic pollutants such as ions of the heavy metal mercury [35].

When it comes to acting in the nanoremediation of pollution caused by dyes, this plant has already provided material for the production of AgNPs capable of performing this task. However, these nanomaterials depended upon direct sunlight or special lamps to perform this action efficiently [36,37]. Some of these AgNPs also exhibited antioxidant potential [37].

The present work is dedicated to the optimization of the Leucena extraction procedure to obtain the highest possible amount of phenolic acids and to the use of this extract to produce biogenic AgNPs that exhibit antioxidant activity and are also able to remediate the pollution caused by dyes without requiring direct exposure to sunlight or to any special lamp or to a specific reduction agent.

## 2. Results

### 2.1. Phenolic Compounds’ Dosage in Crude Extracts Obtained at Different Temperatures

The results of the phenolic compounds’ dosage for the extracts obtained from *L. leucocephala*‘s leaves, stem, and fruits at different temperatures are presented in Table 1. The temperature of 80 °C offered the possibility to increase the extraction of phenolic compounds. The extract from the leaves presented 323.41 ± 0.01 mg of phenolic compounds/g of vegetal material. The stem extract exhibited 835.41 ± 0.04 mg/g, and the one from the fruits presented 183.02 ± 0.16 mg/g.

### 2.2. Phenolic Compounds’ Dosage in Crude Extracts Obtained in Different Time Intervals

The results of the phenolic compounds’ dosage obtained using the extracts from the leaves, stem, and fruits at 80 °C in different time intervals are presented in Table 2. The time interval of 20 min offered the best phenolic extraction when using the leaves (383.95 ± 0.17 mg of phenolic compounds/g of vegetal material) and stem (835.41 ± 0.04 mg of phenolic compounds/g of vegetal material) material. However, for the fruits, the time interval that offered the best result among the ones tested was 30 min (197.58 ± 0.01 mg of phenolic compounds/g of vegetal material).

### 2.3. Nanoremediation Assay with AgNPs Produced Using Different AgNO_3_: Extract Proportions

The potential of AgNPs obtained through the protocols applying different proportions of silver nitrate:extract to promote dye remediation is presented in Table 3. The proportion 1:9 offered the best results with all the dyes and extracts tested.

### 2.4. AgNPs Characterization

The AgNPs synthesized from the Leucena leaf, stem, and fruit extracts were subjected to characterization through UV–vis spectroscopy (300–800 nm). The maximum values of absorbance observed were at 464 nm, 449 nm, and 458 nm, respectively (Figure 2).

Transmission electron microscopy (TEM) analysis revealed different shapes of nanoparticles, but most of them were quasi-spherical AgNPs, presenting different size distribution profiles and consequently different average sizes depending on the plant extract applied in their synthesis. For these quasi-spherical AgNPs, an average diameter of 77.0 ± 19.7 nm was exhibited by the ones synthesized using the extract from the leaves (Figure 3a), 25.5 ± 19.2 nm by the ones produced using stem extract (Figure 3b), and 46.4 ± 16.0 nm by the ones produced using material from the fruits (Figure 3c).

The presence of silver atoms in the nanomaterials was confirmed by electron scattering spectroscopy (EDS), in which signals associated with Ag could be noticed (Figure 4).

Scattering dynamic light (SDL) analysis revealed that the hydrodynamic diameter of the green AgNPs synthesized from the leaves was 181.57 ± 20.80 nm. The value of the polydispersity index was approximately 0.240. Regarding the zeta potential, the average value displayed by the nanomaterial was −40.70 ± 1.41 mV. When it comes to AgNPs produced by applying the stem extract, the hydrodynamic diameter was 222.33 ± 18.41 nm, the polydispersity index was 0.186, and the zeta potential was −42.64 ± 1.53 mV. Regarding the fruit extract-based AgNPs, the hydrodynamic diameter was 440.11 nm ± 15.63, the polydispersity index 0.281, and the zeta potential −28.13 ± 1.09 mV.

X-ray photoelectron spectroscopy (XPS) (Figure 5) revealed signals associated with silver from binding energies of 366.13 eV to 374.28 eV in all three AgNP samples. The presence of carbon and oxygen could also be identified through binding energies from 284.25 to 285.14 and 532.43 to 533.19 eV, respectively.

Through X-ray diffraction (XRD) analysis it was observed that the AgNPs were fcc and crystalline in nature (Figure 6). The mean size of the crystallites calculated using the angle of approximately 38.2 rad from the crystallographic plane (1 1 1) was 9.79 nm for the AgNPs produced using the extract from Leucena’s leaves and 7.83 nm for the AgNPs produced using the extracts obtained from the stem or fruits.

### 2.5. Nanoremediation Assay Using Different Concentrations of AgNPs

The synthesized green AgNPs proved to be able to remediate tartrazine pollution in aqueous samples with different efficiencies; the one synthesized from the leaves was the most efficient (Figure 7).

Regarding the dye methylene blue (Figure 8), the remediation was also performed by the nanomaterials; the AgNP produced using the extract from the leaves was the one that exhibited the best performance.

### 2.6. Dyes’ Adsorption Kinetics

The results of the adsorption kinetics experiment using tartrazine are presented in Figure 9. The best adjustment to the tested models was evidenced for the AgNPs synthesized using the extract from the leaves, which exhibited an R^2^ of 0.995 with regard to the pseudo-second order. The q_e_ for this scenario was 123.05 mg/g of nanomaterial.

The results of methylene blue’s adsorption kinetics are presented in Figure 10. The best adjustment was shown for the AgNPs synthesized using the extract from the leaves, which exhibited an R^2^ of 0.999 with regard to the pseudo-second order. The q_e_ for this scenario was 82.24 mg/g of nanomaterial.

### 2.7. Antioxidant Assay

The results of the antioxidant assay are presented in Table 4. The AgNPs produced using the extract from Leucena’s leaves exhibited antioxidant activity that surpassed that of butylated hydroxytoluene (BHT). The EC_50_ for these nanoparticles was 78.06 µg/mL, and for the BHT, it was 393.79 µg/mL.

## 3. Discussion

Invasive flora species, such as *L. leucocephala*, are considered the second biggest cause of biodiversity plant loss in the world [38]. They may also cause losses in agriculture when they negatively affect crops, impairing seed germination and development. When other plant species that can offer a synergic effect are present in the same environment, such as *Capparis flexuosa* when *L. leucocephala* is being analyzed, the damage to target species can be enhanced in terms of extension [39].

However, Leucena is known to contain a diversity of phenolic compounds. These substances, as previously mentioned, may contribute to the green or biogenic synthesis of metallic nanoparticles once they are able to assist/promote the reduction of metal ions and also act as capping agents [25,26,27,28]. For that reason, the experiments in this work were performed using plant extracts that were optimized to contain the highest possible amount of these substances.

The extractions performed at varying temperatures and times followed by the phenolic compounds’ dosage revealed that 80 °C and 20 min for the leaves and stem and 30 min for the fruits were conditions capable of favoring the obtainment of these substances. The results indicated that the fruits were the part of the plant containing the smallest content of phenolic molecules, which is in accordance with the results obtained by Singsai and coworkers [40].

The AgNPs synthesized from the leaves, stem, and fruits that presented the best remediation potential were characterized. Regarding the AgNPs’ characterization, the absorbance maximum observed through the UV–vis analysis for all the samples was found in the region between 400 and 500 nm. This result is in accordance with literature data once the plasmonic resonance of the silver nanoparticles is responsible for the appearance of the absorbance maximum in this region [41]. However, the value associated with this maximum varied depending on the type of extract used in the synthesis protocol. As can be observed in Figure 3, AgNPs presented a different average size: leaf AgNPs > fruit AgNPs > stem AgNPs; this difference in size influences the UV–vis spectra. The position of the peak associated with the maximum absorbance was shifted to the right part of the graph due to the increase in size. This is referred to as red shift due to the fact that the shifts occurred towards longer wavelengths. The same phenomenon was observed by Dong and coworkers while studying the antibacterial activity of AgNPs of different sizes against *Vibrio natriegens* [42].

The silver atoms’ presence could be also confirmed by the EDS results; a strong signal, for example, between 2.8 keV and 3.4 keV is indicative of silver presence, and it is commonly noticed in studies that perform the characterization of this type of nanoparticle [43].

The TEM images made it possible to access the shape and size distribution of the AgNPs produced. Different shapes of these particles have already been reported, such as triangular, pentagonal, rod, and hexagonal, but the spherical one is the most commonly reported [44]. While some variations in this spherical shape are noticed, they are not sufficient to make it possible to classify the AgNPs as presenting another type of shape; the nanomaterial is considered to present a quasi-spherical shape, such as that of the nanoparticles produced by Ankudze and coworkers [45].

The polydispersity index is capable of indicating the heterogeneity of sizes present in a sample. Nanomaterials can present different sizes in the same sample, or they may also agglomerate/aggregate in solution to produce larger materials, for example. The values obtained for the AgNPs produced from leaf, stem, or fruit extracts are below 0.5; this result is indicative of monodispersed samples [46]. Regarding the hydrodynamic diameter, as expected it is much larger than the particle size observed using TEM due to the fact that around the AgNP there is a surface layer which participates in the stabilization of the nanostructure [47].

XPS analysis is important and allows studies of AgNPs’ surface composition [48]. The results obtained made it possible to identify signals associated with silver from binding energies of 366.13 eV to 374.28 eV on the surface of all three AgNP samples. These signals refer to Ag’s 3d core levels; the first (Ag 3d _5/2_) is associated with the binding energy of metallic silver and the second one (Ag 3d _3/2_) with the binding energy related to the formation of AgNPs. The signals associated with these core levels might vary, shifting to higher binding energies, but for nanoparticles smaller than 10 nm, which is not the case in this study. The signals obtained are in good agreement with the ones obtained in studies in which other AgNPs were produced [49,50,51,52,53,54]. The difference between the two Ag peaks (Ag 3d _5/2_ and Ag 3d _3/2_) was about 8 eV, which is consistent with the pure metallic state of silver [55]. The presence of carbon and oxygen atoms could also be identified through binding energies from 284.25 to 285.14 and 532.43 to 533.19 eV, respectively [49]. These results were expected due to the fact that the extracts were obtained in a manner that enhanced the amount of phenolic compounds present, and these substances were rich in these elements [56].

XRD analysis allowed the identification of AgNPs as fcc and crystalline; the peaks presenting 2θ values of 38.20 to 38.32, 44.40 to 46.36, 64.64 to 67.72, and 76.96 to 77.54 could be assigned to the crystallographic planes (1 1 1), (2 0 0), (2 2 0), and (3 1 1), respectively. These results are in agreement with the ones presented in the literature regarding green AgNPs [57,58] and chemical ones [59]. The other peaks are associated with the bio-organic phase on AgNPs’ surface [57,58], and consequently, a different pattern is displayed depending on the plant part used to obtain the extract for synthesis. This is due to differences in the composition of this phase in each sample. The mean size of the crystallites presented the highest value in the AgNPs produced from Leucena’s leaves (9.79 nm), which was also the largest nanoparticle (Figure 3). However, the size of the crystallites is not necessarily the same as the size of the nanoparticles, due to the fact that the nanostructure can be polycrystalline. In this work, the nanoparticles presented a crystallite size smaller than the particle size; in polycrystalline nanoparticles, such as those synthesized from Leucena, this occurs because their surface is composed of various tiny areas presenting a well-organized structure but each one of those areas is separated from the others by grain boundaries [60].

When it comes to the remediation assays, the nanomaterials performed the remediation of aqueous samples containing tartrazine and methylene blue, although the efficiency of this process varied depending on the dye and on the source of the extract used for synthesis. The occurrence of remediation is a desirable aspect because these dyes are very harmful to living beings; tartrazine, for example, is largely used in the food industry when the intention is to produce a yellow color. However, it is reported as being able to exert toxic effects even at a dose considered to be the one that humans are allowed to consume on a daily basis. It can establish undesirable molecular interactions that threaten living beings’ health and survival [61]. Over human gastric cells and fibroblasts, for example, it can act by producing a cytotoxic effect that increases in a dose-dependent manner. In vegetal species that may be in contact with the substance when it is discharged in the environment as a pollutant, the negative effects are also present. The contaminant was proven to induce a mutagenic effect on *Allium cepa* [62]. If the industrial wastewater containing tartrazine reaches lakes or rivers, for example, fish species will also be in danger. The dye proved to be neurotoxic in doses such as 50 mg/L; it can disturb antioxidant systems causing, as a consequence, oxidative stress. Consequently, the cells are damaged in structure but also impaired in their maintenance of their regular functions. Undesirable neuro-biochemical changes were already observed in zebrafish embryos that suffered extensive cell death through mitochondria-mediated apoptosis [63]. In fact, the aquatic toxicity can be a consequence not only of tartrazine’s presence, but also of its by-products. In *Danio rerio* these substances impacted the reproductive processes, accelerating reproduction [61].

Tartrazine is an azo-dye like methylene blue, which is also a toxic substance to living beings. Algae such as *Chlorella vulgaris* and *Spirulina platensis* can suffer the negative consequences of exposure. A dose of 100 mg/L could induce, after 96 h of exposure, a reduction in the chlorophyll and carotenoid content in both species; however, *C. vulgaris* was more affected than *S. platensis*. The protein level in both species was reduced even when the assays were performed using a dye concentration of 20 mg/L, and as the concentration increased, the reduction was also more drastic [64]. Humans can also experience negative effects as a consequence of exposure to this dye; these effects include vomiting, tissue necrosis, cyanosis, and even shock [65]. Therefore, it is necessary to develop and apply sustainable methods to remediate the pollution caused by synthetic dyes such as the organic environmental contaminants addressed in this work.

The aqueous samples containing methylene blue or tartrazine could be remediated without performing direct exposure to sunlight or special lamps such as xenon ones. It was also not necessary to add NaBH_4_ or other reducing agents. So, although further investigation is necessary to provide the details associated with the mechanism involved in remediation, it is possible to present preliminary data related to the kinetics of adsorption.

Silver nanoparticles have already been proven to promote pollution management through the contaminants’ adsorption or degradation; the degradation can be performed through photocatalysis (a degradation mechanism that depends on light) or chemical degradation (a light-independent mechanism of degradation). However, to perform the last mentioned type of degradation, specific reduction agents are commonly added, such as H_2_O_2_, NaBH_4_, and citrate for example [66]. As previously mentioned, in this study the samples could remediate aqueous samples containing dyes in a manner that was independent of the direct incidence of light, and no reducing agent was added during the process, making adsorption the mechanism of action.

Analyzing the kinetics of tartrazine and methylene blue’s remediation, the AgNPs from the leaves were the ones that offered the best adjustment to the first- and second-order kinetics, and they adjusted better to the latter. The q_e_ for these AgNPs surpassed the one observed by Bathol and coworkers (approximately 97.1 mg/g) [67] and the one from Bayik and Baykal (1.15 mg/g) [68]. According to pseudo-second order kinetics, a chemical adsorption (involving electron transfer or sharing) guides the sorption [69].

Regarding the other AgNPs produced, their kinetics still need to continue to be investigated using other models, such as the Elovich and Weber–Morris models [70]. Isotherm models and studies involving the influence of pH, of the AgNP concentration and of temperature in adsorption processes, involving all the nanoparticles synthesized (such as the one performed by Pandian and coworkers [71]), are also future perspectives that will receive attention.

When it comes to the difference in efficiency observed among the nanoparticles in remediating the pollution caused by dyes, the contact surface needs to receive attention. Methylene blue is a cationic dye; the zeta potential analysis performed in samples of AgNPs in deionized water (pH 5.8) indicated a negative surface charge, which may have favored charge interaction during adsorption.

When it comes to tartrazine, an anionic dye, the effect on adsorption, at first sight, would be expected to be the opposite of the one observed for methylene blue due to repulsion. However, the AgNPs could deal with both dyes. Analyzing the details associated with the experiments, they were all conducted at room temperature, but the pH was not adjusted prior to remediation or adsorption analysis. By verifying the pH value, it was observed that tartrazine–leaf AgNPs gave a pH of 4.9 and methylene blue–leaf AgNPs gave 5.9. This difference in pH could have influenced the protonation of the AgNP surface groups present in the phenolic acids used during synthesis/capping. Gallic acid, which is a phenolic acid, enhances its deprotonation as pH increases, becoming a more negative structure at higher pHs [72]. The pH below the acid dissociation constant of tartrazine (pKa = 9.4) may favor this adsorption as the dye also tends to diminish its negative charge [73,74]. Tartrazine was already proven to interact efficiently with carbon-rich structures. Silver nanoparticles functionalized with dodecanethiol by interactions involving the thiol group exhibited carbon on their surface; these modified AgNPs exhibited a high rate of tartrazine recovery from food samples through single-drop microextraction [75]. Consequently, at a lower pH, the negative surface charge of AgNPs and the one from the dye diminish, contributing to a decrease in repulsion between them. This can favor other interactions between AgNPs and the dye such as those involving carbon, which was shown by XPS analysis to be on the nanomaterial’s surface. The AgNPs from the leaves presented the strongest signal associated with carbon in XPS when compared to AgNPs from the stem and fruits (Figure 5). This presence could have favored the interaction of carbon with the dye tartrazine. It is important to highlight the fact that this nanoparticle is also the one presenting a larger size and exhibiting a larger surface area, which can also favor adsorption.

Methylene blue’s remediation took place at a higher pH than that of tartrazine. The mentioned behavior of phenolic acids regarding protonation/deprotonation could have favored a more negative surface charge and consequently the interaction with the cationic dye. The zeta potential that was analyzed at a pH near to the one from the methylene blue–AgNP solution was negative for all the nanoparticles. The AgNPs obtained from the extract of the fruits exhibited the smallest negative surface charge and the less efficient performance; the zeta potential from the AgNPs from the leaves and stem was similar, but the leaves exhibited a larger surface area and consequently a larger efficiency in performance during remediation. As a future perspective, the zeta potential of the nanoparticles will also be determined at a pH of 4.9 to verify whether the negative surface charge diminishes, contributing to a decrease in repulsion between tartrazine and the AgNPs and favoring carbon–dye interactions during adsorption.

The leaves of *L. leucocephala* have already been studied as raw material for the synthesis of green AgNPs that exhibit the potential to remediate dye pollution. By carrying out a different synthesis protocol time (30 min) with a different proportion of silver nitrate to extract (0.5 g of AgNO_3_ for 50 mL of extract), Hazim and collaborators obtained a different silver nanoparticle. Presenting an average size of 35 nm (smaller than the ones synthesized in this work) and depending on a reactor with a 200 W visible light source (produced by a Xenon lamp), the nanoparticle, at 1000 ppm, could remove approximately 78% of the synthetic dye used in the experiments [36]. With a different ratio of Leucena leaves to distilled water (5 g to 1000 mL), extracted overnight, Raju and collaborators also obtained a plant extract for AgNPs synthesis. In a 1:1 ratio of salt containing silver:extract, nanomaterials of different sizes were generated in two main diameter ranges: 15–20 nm and 30–35 nm. The nanomaterials relied on direct exposure to sunlight to deal with Red m5b dye pollution through photocatalysis [37]. The nanoparticle produced in this work promoted the remediation of the pollution caused by tartrazine and methylene blue with higher efficiency and without requiring direct light exposure, which is an advantageous feature when large scale application is a future goal, favoring a process with lower costs that is easily performed.

Regarding the antioxidant activity exhibited by the AgNPs, the extracts obtained from the seeds of Leucena have already been reported as being able to allow the biogenic synthesis of AgNPs; these particles presented antimicrobial and antioxidant activities and selective optical sensing towards Fe^3+^ [47]. From the leaves of Leucena, synthesized AgNPs have already been produced that exhibit antioxidant activity [37]; however, the AgNPs synthesized in this work exhibited a superior antioxidant potential. This activity is of special interest in the fight against free radicals: the ones produced as a consequence of metabolism as well as the ones that are a consequence of exposure to environmental contamination, immune response, ionizing radiation, and so on [76]. These structures are associated with a large array of diseases [77], such as Parkinson’s and Alzheimer’s disorders [78], asthma [79], atherosclerosis [80], and cancer [81], due to the fact that they, as reactive substances, might damage cell structure and impair biomolecules’ regular functioning. Antioxidants are used to mitigate free radical production and action, contributing to longevity in a healthy manner [77]. Nanoparticles can act as nano-antioxidants, offering advantages such as the possibility to be designed to perform targeted delivery and controlled release as well as to have increased bioavailability [77], which is especially useful in the medical field. They can favor, for example, cancer treatment [82] and wound healing [83]. Therefore, the AgNPs synthesized in this work using the leaves of Leucena and those that exhibited this potential can not only be used in remediation protocols; they can also undergo cytotoxicity assays to assess their potential to be used as antioxidant structures in the aforementioned applications.

The previously mentioned AgNPs produced by Raju and collaborators using the extract obtained from the leaves of Leucena exhibited EC_50_ 240.70 µg/mL [37], which was greater than the value presented by the AgNPs produced here using the extract from the leaves of this species. These AgNPs synthesized in this work, besides being capable of performing nanoremediation, have greater antioxidant potential than the ones produced by Raju and coworkers: they require a lower concentration to provide the effect of 50% discoloration of the DPPH solution (EC_50_ = 78.06 µg/mL) and offer an interesting potential to be further investigated.

Considering the fact that from the AgNPs synthesized in this work only the one obtained using the extract from the Leucena’s leaves exhibited antioxidant activity that surpassed the one from BHT, their surface deserves to receive attention once more. As previously mentioned, the extracts used were the ones that presented the higher phenolic acid content. Although the stem extract presented a higher content of these substances when compared to the leaf extracts, XPS analysis revealed that the signals of carbon and oxygen in the surface of the AgNPs from the leaves were higher. These atoms are largely present in phenolic acids.

Abdel-Aty and coworkers have produced green AgNPs from a saw palmetto seed extract that was optimized to contain phenolics. The nanoparticles, presenting a zetapotential of −32.8 mV, exhibited antioxidant activity that surpassed the one exhibited by the crude extract and by the antioxidant standard Trolox. Phenolic acids, besides being efficient substances in the synthesis/capping of AgNPs [27], are also substances with a relevant antioxidant activity [84,85]. So, their existence in larger amounts in the surface of the AgNPs obtained using the extract from the leaves favored the exhibition of antioxidant activity.

Consequently, as a future perspective, as previously mentioned, the AgNPs will continue to be studied regarding the remediation process mechanism. Different kinetic models will be explored and the influence of variables such as pH, concentration of adsorbent, and temperature in adsorption, will receive attention. The influence of pH in dye interaction with nanoparticles will also be further investigated.

In addition, it is also important to explore how these nanomaterials can be efficiently recovered after the end of a remediation cycle and the possibility to reuse them. Regarding the recovery process, bio-based adsorbents are an interesting solution; a chitosan-based cryogel, for example, could remove up to 96.5% of AgNPs from water samples [86].

Another important aspect to explore is cytotoxicity. In particular, the AgNPs produced using the extract from the leaves of Leucena need to be analyzed due to their antioxidant potential, and the cytotoxicity aspects need to be verified in vitro and in vivo.

## 4. Materials and Methods

### 4.1. Preparation of Plant Material

The plant material was collected in the city of Divinópolis-MG, Brazil, at the following geographic coordinates: 20°8′7″ S and 44°53′49″ W. The voucher specimen was deposited in EPAMIG’s herbarium under the number PAMG 58919, after the botanical identification carried out by the biologist Andréia Fonseca Silva. The collected material was cleaned, and the leaves, stem, and fruits (containing seeds) were separated and dried in an oven (Ethik Technology; Vargem Grande Paulista, Brazil) with air circulation at 40 °C until their mass stabilized. The dried material was pulverized in a knife mill (Tecnal; Piracicaba, Brazil) and stored in order to be submitted to the process of crude extract obtainment.

### 4.2. Crude Extract Obtainment at Different Temperatures

The crude aqueous extracts using the material from the leaves, stem, and fruits were obtained in ultrasonic bath Soniclean 2PS (Sanders Brasil; Santa Rita do Sapucaí, Brazil) using 10 g of plant material and 100 mL of deionized water for 30 min at different temperatures (20 °C, 40 °C, 60 °C, 80 °C, and 100 °C). Subsequently, the crude extracts were filtered and stored in amber containers under refrigeration at −4 °C.

### 4.3. Phenolic Compounds’ Dosage

The dosage of phenolic compounds was carried out using the Folin–Ciocalteu colorimetric method [87] with adaptations. Gallic acid 0.2 mg/mL (Sigma Aldrich; Duque de Caxias, Brazil) was used as a standard in the construction of the analytical curve (5, 10, 20, 30, and 40 µg/mL). The samples, in triplicate, were added to water to complete the volume of 250 μL; then, 2250 μL of the reactive mixture (Folin–Ciocalteu) was added. After 30 min, the absorbance was determined at 750 nm in a spectrophotometer Genesys 10uv (Thermo Fischer; São Paulo, Brazil). The values were converted to indicate mg of phenolics/g of plant material.

### 4.4. Crude Extracts Obtainment in Different Time Intervals

The crude aqueous extracts from Leucena’s leaves, stem, and fruits were obtained in ultrasonic bath Soniclean 2PS (Sanders Brasil; Santa Rita do Sapucaí, Brazil) using 10 g of plant material and 100 mL of deionized water at 80 °C (optimized temperature) in time intervals of 10, 20, 30, 40, 50, or 60 min. Subsequently, the crude extracts were filtered and stored in amber containers under refrigeration at −4 °C. The phenolic compounds’ dosage was performed as described in Section 4.3.

### 4.5. AgNPs Synthesis

An amount of 1 mM aqueous solution of silver nitrate (AgNO_3_—Sigma Aldrich; Duque de Caxias, Brazil) was prepared. It was combined with the crude extracts presenting optimized phenolic acid content under agitation in a magnetic stirrer IKA C-MAG HS 7 (IKA; Campinas, Brazil), in the absence of light. Different proportions of silver nitrate:extract *v*/*v* were used: 1:1, 1:2, 1:3; 1:4; 1:5; 1:6; 1:7; 1:8; 1:9; and 1:10. The color was verified every 20 min until it changed to dark brown. The reaction products were centrifuged (NT800—Nova Técnica; Piracicaba, Brazil) at 14,000 rpm for 20 min, and the pellets obtained were washed 3 times with deionized water to remove residues. The AgNPs were dried at 70 °C.

### 4.6. Nanoremediation Assay

Solutions of the tartrazine (popularly applied in the food industry) and methylene blue (popularly applied in the textile industry) dyes were prepared at a concentration of 16 mg/L to allow a final test concentration of 10 mg/L [22] after dilution in microplate tests. This assay was performed in 96-well plates, in triplicate, to test 5 different concentrations. The triplicates pipetted for each test were (1) water; (2) dye (always with a final concentration of 10 mg/L); (3) NPs at 1000 ppm in water; (4) NPs at 1000 ppm and dye; (5) NPs at 500 ppm in water; (6) NPs at 500 ppm and dye; (7) NPs at 250 ppm in water; (8) NPs at 250 ppm and dye; (9) NPs at 125 ppm in water; (10) NPs at 125 ppm and dye; (11) NPs at 62.5 ppm in water; and (12) NPs at 62.5 ppm and dye. The maximum concentration tested was the same one used in the remediation assays performed by Foster and coworkers [88], and the others were obtained through serial dilution with 2 as the dilution factor.

An absorbance reading was performed every 30 min for 2.5 h at 427 nm (tartrazine) or 665 nm (for methylene blue). The results were analyzed by two-way ANOVA with Tukey’s post-test, using Prism 7.00 software (GraphPad Software; Boston, MA, USA) [89].

### 4.7. AgNPs’ Characterization

The amount of 50 µL of a 1000 ppm vial containing the synthesized AgNPs that displayed the best remediation potential for each plant part were diluted in 950 µL of deionized water and were subjected to characterization through UV–vis spectroscopy (200–800 nm), EDS, SDL, XPS, and XRD. Based on the XRD results, the mean size of the crystallites from each sample of AgNPs was calculated using the Debye–Scherrer equation (Equation (1)) [57]:D = 0.94 λ/β cosθ (1)

The zeta potential (Litesizer DLS 500—Anton Paar; Graz, Austria) and TEM images (Tecnai^TM^ G^2^ Spirit Biotwin 120 kV—FEI Company; Hillsboro, OR, USA) of the AgNPs were also obtained. Regarding the TEM analysis, the size of the particles present in the images were measured using ImageJ 1.54 g (ImageJ, Bethesda, MD, USA) to access the size distribution profile.

### 4.8. Dyes’ Adsorption Kinetics

The mixture of AgNPs (25 mg) from each plant part and each dye (3 mg) in water to reach the final volume of 2.5 mL was stirred using a magnetic stirrer at room temperature for 30 min to allow an initial equilibrium. After this time lapse, the solutions were filtered, and the supernatants were analyzed using a UV–vis spectrophotometer at the wavelength previously mentioned for each dye to assess the dye concentration in the residual solution. Measurements were performed every 30 min for 2.5 h. The rates at which the dyes adsorbed onto the AgNPs were analyzed by applying pseudo-first and pseudo-second order equations [65].

OriginPro 10.1.0.170 (OriginLab, Northampton, MA, USA) was used in this analysis. For the pseudo-first order, Equation (2) was used, and for the pseudo-second order, Equation (3) was used.
q_t_ = q_e_(1 − e^−kt^)(2)
q_t_ = (kq_e_^2^t)/(1 + kq_e_t)(3)

In the previously presented equations, q_t_ is the adsorption capacities at time t, q_e_ refers to the adsorption capacity at equilibrium, k is the rate constant, and t the time [90].

### 4.9. Antioxidant Assay

The AgNPs also had their antioxidant potential analyzed. 1,1-Diphenyl-2-picrylhydrazyl (DPPH) is a free radical used to evaluate antioxidant activity. Determination of antioxidant activity by the DPPH method [91] was adapted for use with microplates. A solution of DPPH (0.002% *w*/*v*) was prepared in ethanol. The 75 μL volume of samples or the standard (BHT) was added to each of the wells of the 96-well microplate, which contained 150 μL of the DPPH solution. Samples were prepared in triplicate for each of the five concentrations used (1, 10, 100, 250, and 500 μg/mL). The plate was then covered and left in the dark at room temperature (25 °C). After 30 min, the absorbance at 517 nm was determined using a spectrophotometer. The pure solvent was used as a blank for the experiment. The percentage of DPPH inhibition was calculated using the following equation (Equation (4)) by Burda and Oleszek [92]:% DPPH inhibition = [1 − (Aa/Ab)] × 100(4)

The Aa refers to the absorbance of the sample and Ab to the absorbance of the DPPH solution. The calculation of EC_50_ (effective concentration to discolor 50% of the DPPH solution) was carried out using the probits method of analysis [93].

## 5. Conclusions

It was possible to synthesize AgNPs using the aqueous extracts obtained at the temperature and in the time interval that could offer an increased concentration of phenolic compounds. The extracts from the leaves, stem, and fruits of *L. leucocephala* all proved that they were able to be the bio-based raw material for the synthesis of the green nanomaterial containing silver. The quasi-spherical polycrystalline nanoparticles presented a negative surface charge and a polydispersity index inferior to 0.3. They presented different average sizes, as visualized through TEM. The presence of silver atoms could be confirmed by EDS, and through XPS analysis, the presence of silver, carbon, and oxygen on the surface could be observed.

Regarding the activities exhibited by nanoparticles, all three biogenic AgNPs could remediate, through adsorption, the pollution caused by the dyes methylene blue and tartrazine using a mechanism that was independent of direct light incidence or specific reduction agent addition. However, the ones synthesized using the extract from the leaves could do it in a more efficient manner. These AgNPs exhibited larger size and, consequently, a larger surface area; they also presented stronger signals associated with the elements present on the AgNPs’ surface (Ag, C, and O). A difference in the pH of the AgNP–dye mixture during the remediation and adsorption kinetics assays may have influenced the interactions involved in the adsorption of methylene blue (a positively charged dye) and tartrazine (a dye that presents negatively charged groups). It is hypothesized that in the first scenario the methylene blue–AgNP association is maintained due to electrostatic interactions (pH 5.9). At a lower pH of 4.9, however, the negative charges may be attenuated, and the carbon-rich structures present in the AgNPs’ surface may drive the AgNP–tartrazine interactions. These AgNPs (from Leucena’s leaves) were also the ones that adjusted better to pseudo-second order adsorption kinetics.

The AgNPs produced using the extract from the leaves of Leucena were also the only nanoparticle sample exhibiting an antioxidant potential capable of surpassing the one exhibited by BHT. As AgNPs from the leaves of *L. leucocephala* had already been produced and tested to perform the functions analyzed in this work, the results could be compared with the ones from the literature. It was possible to conclude that in both functions the AgNPs produced in this work offered better results, as previously discussed in detail.

Therefore, this study offers a protocol to obtain, from the extract obtained using the leaves of an invasive species (in a simple and low-cost manner), AgNPs that can be used to efficiently remediate the organic environmental pollution caused by the dyes methylene blue and tartrazine, independently of direct light incidence or the presence of reducing agents, which favors a future scale-up to treat contaminated wastewater. By focusing attention on the phenolic content in plant extracts, it was possible to produce AgNPs that not only promote the adsorption of dyes presenting better q_e_ than the other previously green-synthesized ones, but that also exhibit more efficient antioxidant activity than the green AgNPs in the references mentioned in this study.

## 6. Patents

The work reported in this manuscript resulted in a patent deposit in Brazil. The INPI deposit number is BR 10 2024 001249 6.

## Figures and Tables

**Figure 1 ijms-25-03993-f001:**
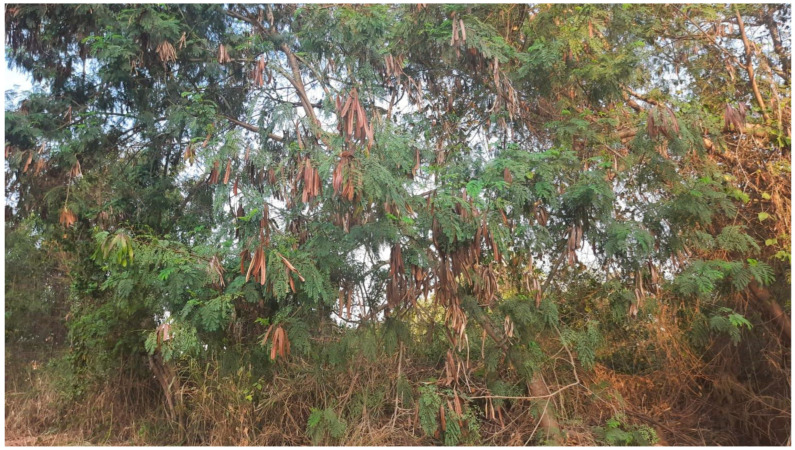
*Leucaena leucocephala*.

**Figure 2 ijms-25-03993-f002:**
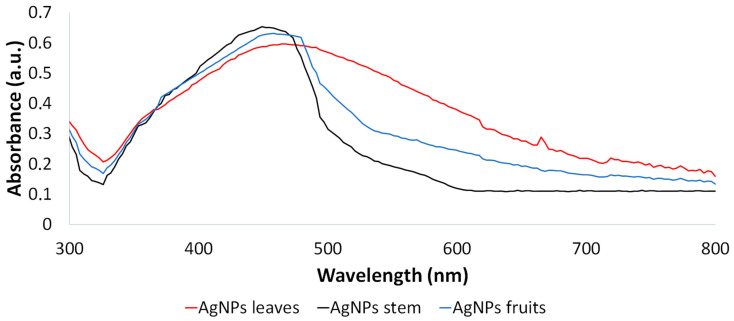
UV–vis spectroscopy of AgNPs synthesized from Leucena’s leaf, stem, and fruit extracts.

**Figure 3 ijms-25-03993-f003:**
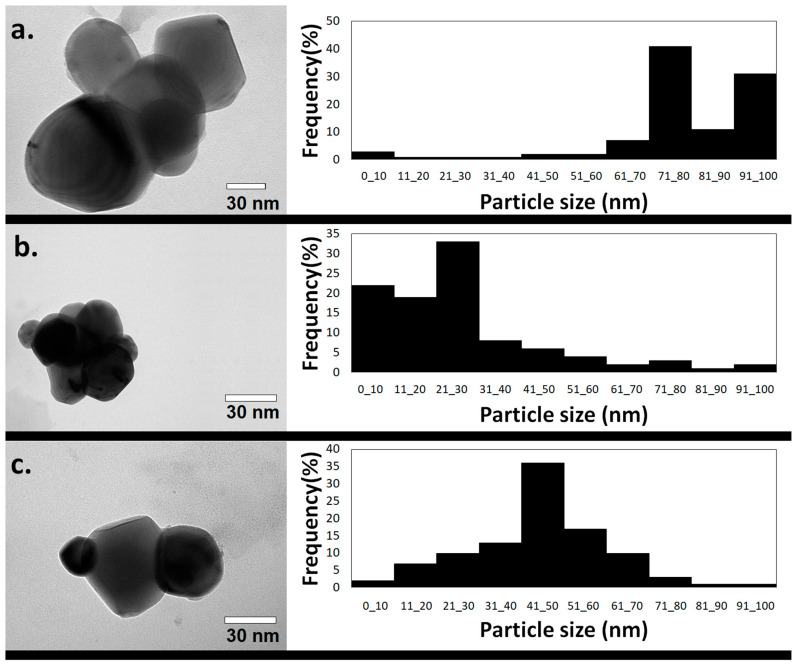
TEM image and size distribution profile of green AgNPs synthesized applying the extracts from Leucena’s (**a**) leaves, (**b**) stem, and (**c**) fruits.

**Figure 4 ijms-25-03993-f004:**
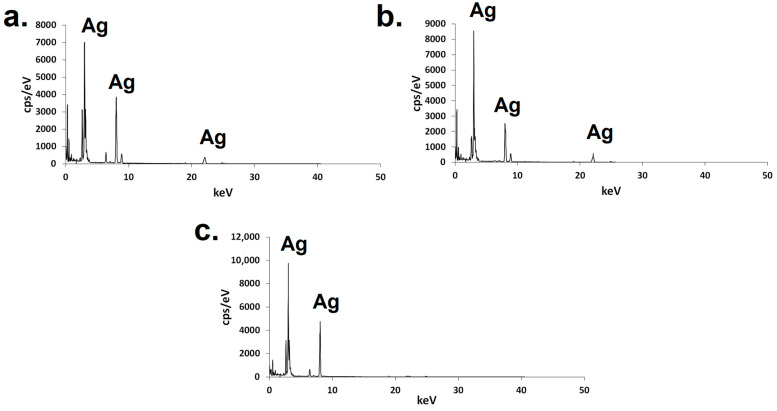
EDS of green AgNPs synthesized using the extracts from Leucena’s (**a**) leaves, (**b**) stem, or (**c**) fruits.

**Figure 5 ijms-25-03993-f005:**
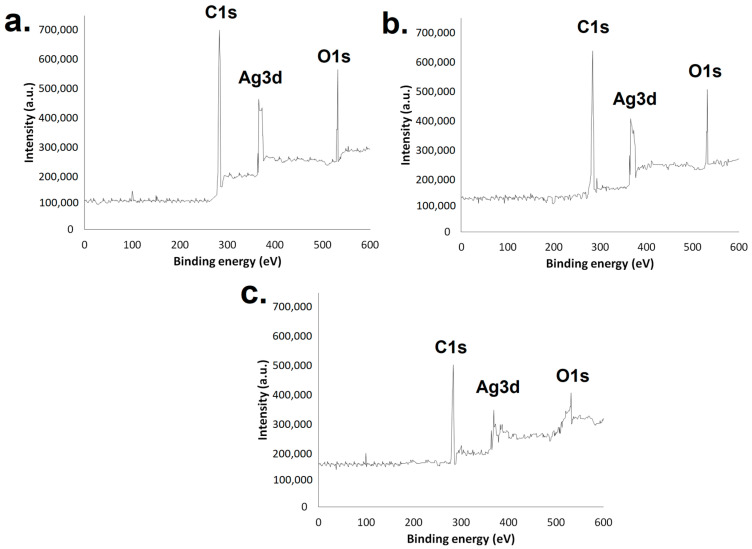
XPS of green AgNPs synthesized using the extracts from (**a**) leaves, (**b**) stem, and (**c**) fruits.

**Figure 6 ijms-25-03993-f006:**
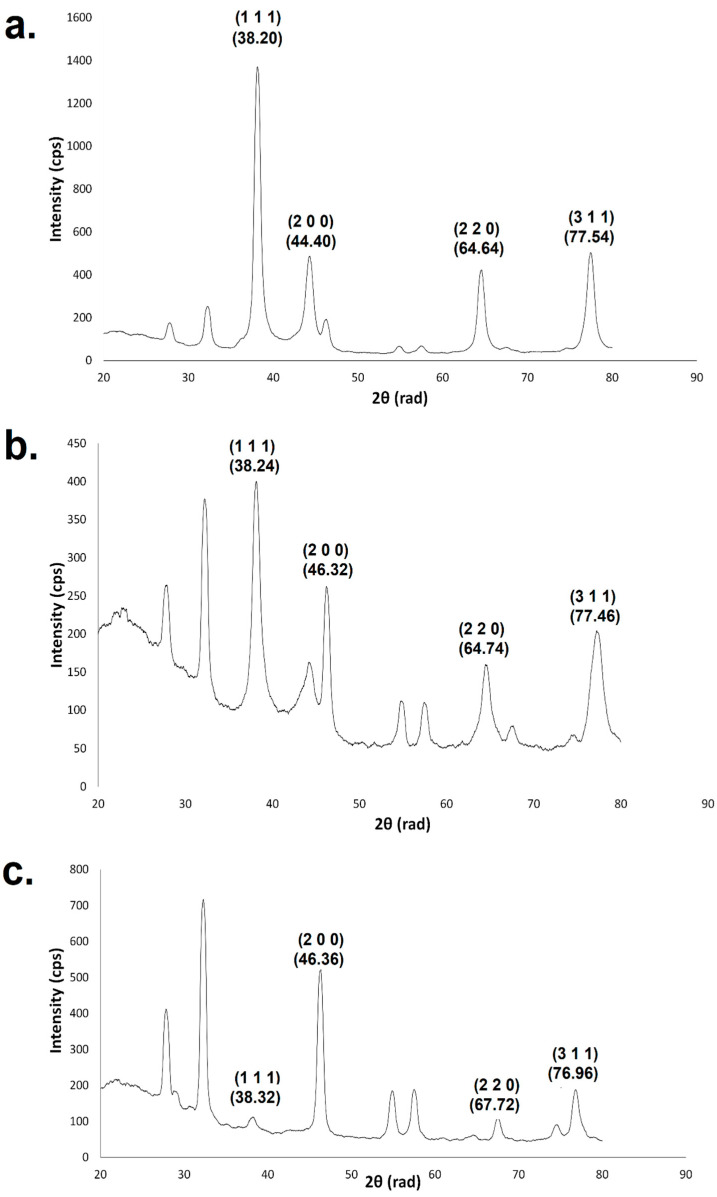
XRD of green AgNPs synthesized using the extract from (**a**) leaves, (**b**) stem, and (**c**) fruits.

**Figure 7 ijms-25-03993-f007:**
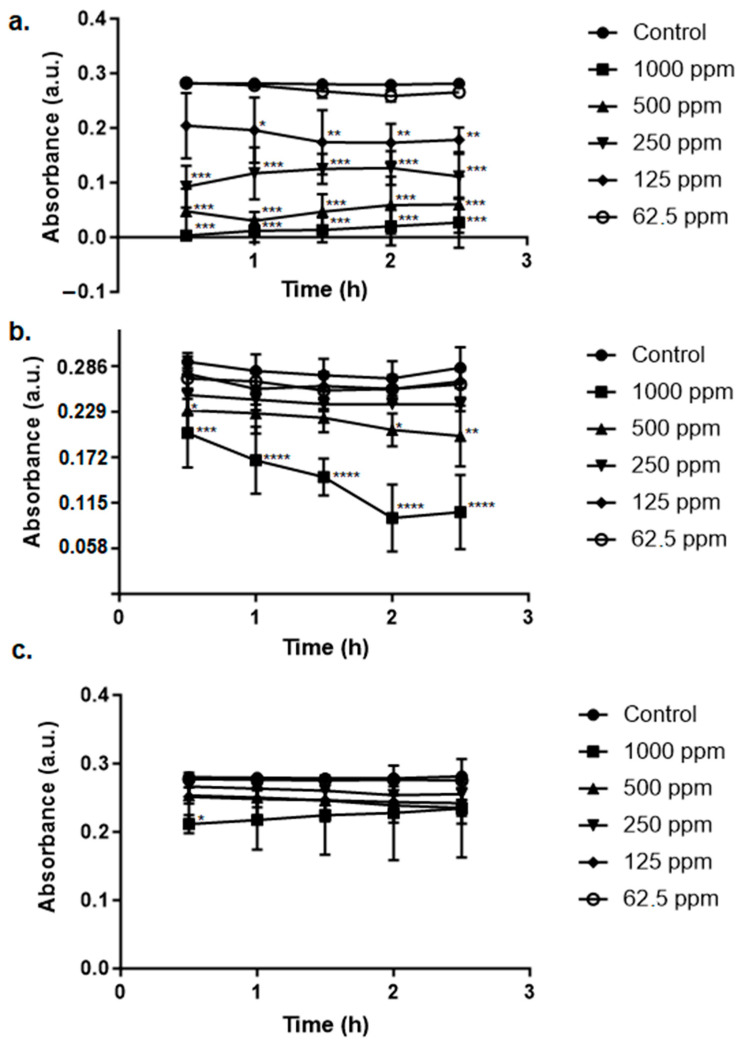
Remediation of tartrazine’s associated pollution by the green AgNPs synthesized using Leucena’s (**a**) leaves, (**b**) stem, and (**c**) fruits at different concentrations. Control is the test with 0 ppm of AgNPs. Statistically significant results compared to the control group are highlighted using asterisk(s) (* *p* < 0.05; ** *p* < 0.005; *** *p* < 0.001; **** *p* < 0.0005).

**Figure 8 ijms-25-03993-f008:**
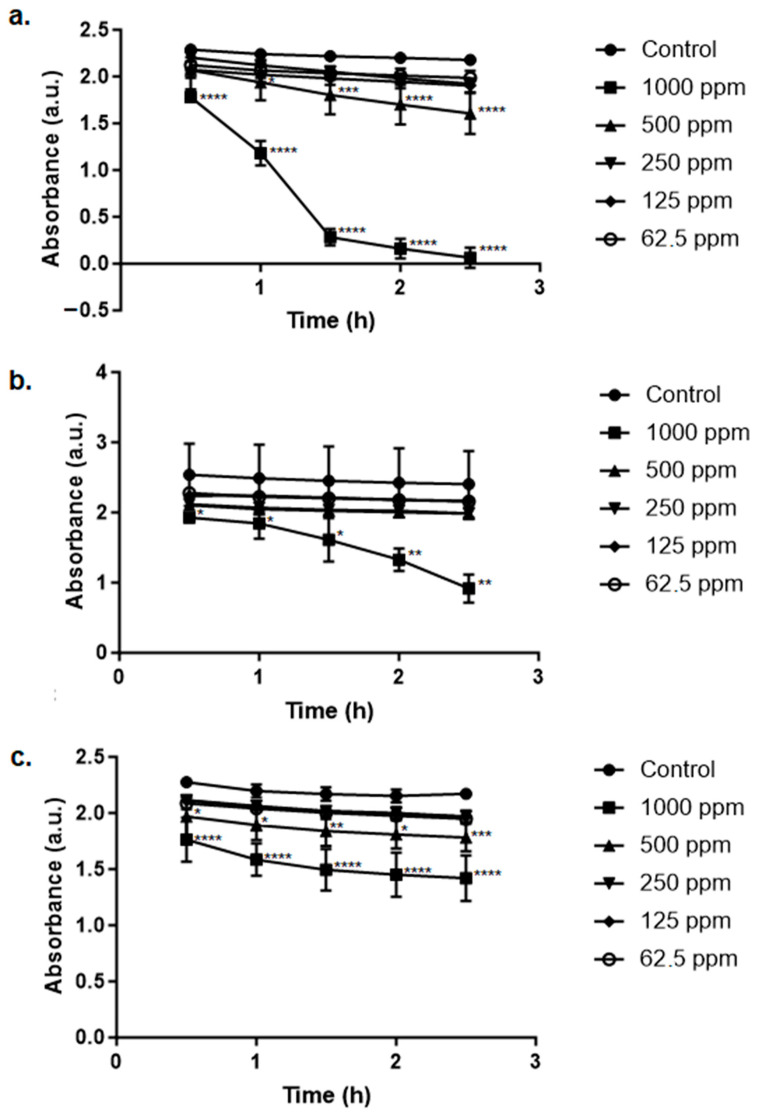
Remediation of methylene blue’s associated pollution by the green AgNPs synthesized using Leucena’s (**a**) leaves, (**b**) stem, and (**c**) fruits at different concentrations. Control is the test with 0 ppm of AgNPs. Statistically significant results compared to the control group are highlighted using asterisk(s) (* *p* < 0.05; ** *p* < 0.005; *** *p* < 0.001; **** *p* < 0.0005).

**Figure 9 ijms-25-03993-f009:**
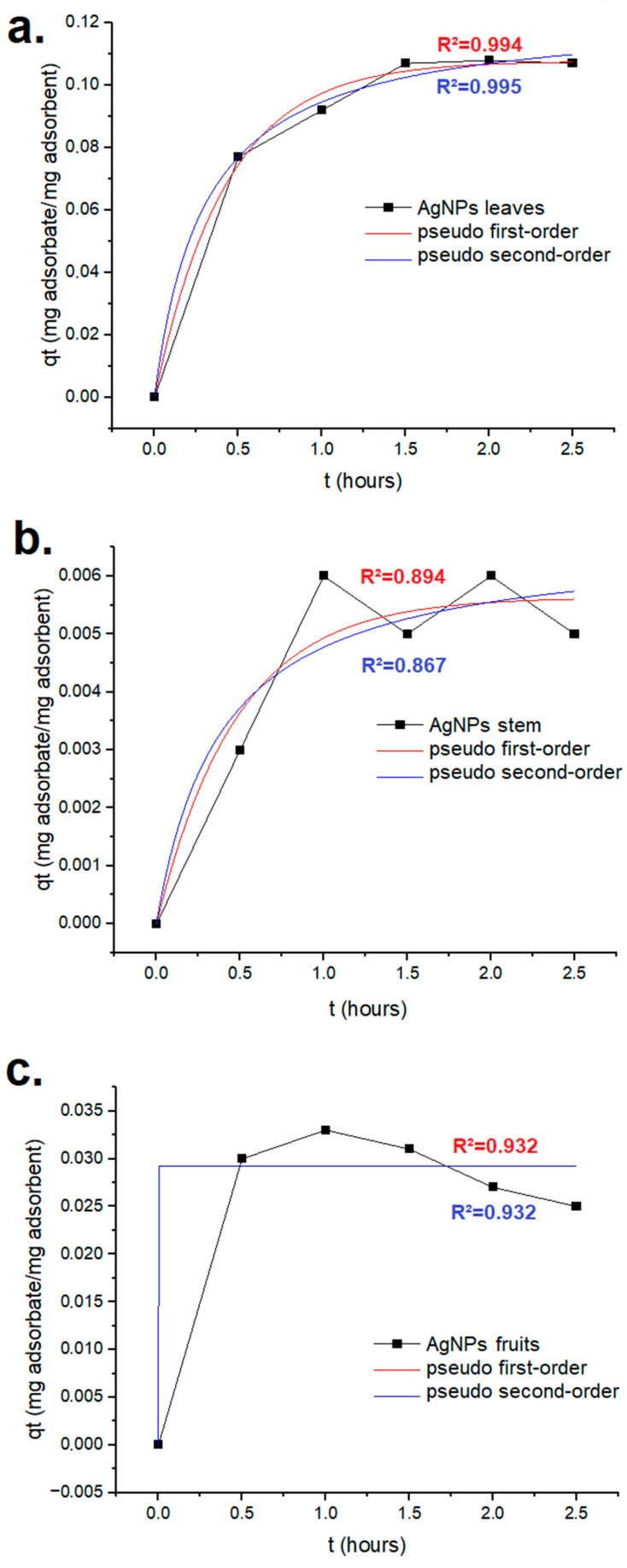
Tartrazine’s adsorption kinetics for the green AgNPs synthesized using Leucena’s (**a**) leaves, (**b**) stem, and (**c**) fruits.

**Figure 10 ijms-25-03993-f010:**
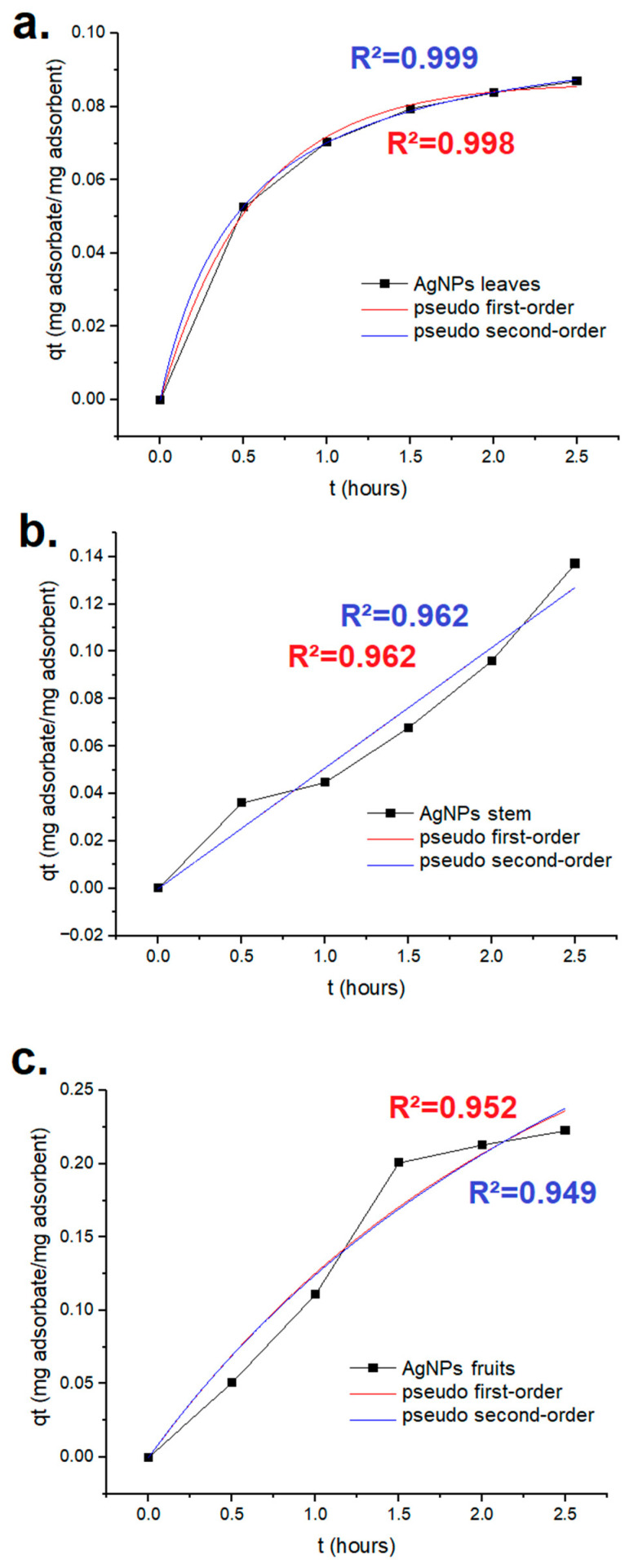
Methylene blue’s adsorption kinetics for the green AgNPs synthesized using Leucena’s (**a**) leaves, (**b**) stem, and (**c**) fruits.

**Table 1 ijms-25-03993-t001:** Phenolic contents of extracts obtained at different temperatures.

Temperatures	Phenolic Compounds’ Dosage (mg/g of Vegetal Material)		
	Leaves	Stem	Fruits
20 °C	239.97 ± 0.03	429.75 ± 0.01	122.29 ± 0.07
40 °C	242.73 ± 0.04	527.89 ± 0.05	130.27 ± 0.02
60 °C	248.00 ± 0.04	795.84 ± 0.07	168.97 ± 0.05
80 °C	323.41 ± 0.01	835.41 ± 0.04	183.02 ± 0.16
100 °C	257.43 ± 0.04	737.89 ± 0.01	152.48 ± 0.07

**Table 2 ijms-25-03993-t002:** Phenolic contents of extracts obtained in different time intervals at 80 °C.

Time Intervals (min)	Phenolic Compounds’ Dosage (mg/g of Vegetal Material)		
	Leaves	Stem	Fruits
10	289.72 ± 0.01	715.67 ± 0.03	164.25 ± 0.02
20	383.95 ± 0.17	835.41 ± 0.04	183.02 ± 0.08
30	379.01 ± 0.11	833.48 ± 0.11	197.58 ± 0.01
40	382.15 ± 0.09	829.28 ± 0.07	191.25 ± 0.02
50	384.15 ± 0.22	834.15 ± 0.05	193.45 ± 0.17
60	383.99 ± 0.11	832.59 ± 0.10	183.19 ± 0.08

**Table 3 ijms-25-03993-t003:** Percentage of dye remediation obtained for the AgNPs.

Proportion Silver Nitrate:Extract	Methylene Blue			Tartrazine		
	Leaves	Stem	Fruits	Leaves	Stem	Fruits
1:1	55.5%	4.25%	9.99%	20.17%	9.15%	10.03%
1:2	57.89%	9.55%	12.25%	27.89%	19.84%	10.57%
1:3	64.61%	10.79%	15.48%	34.15%	22.59%	10.89%
1:4	67.79%	11.27%	17.86%	34.78%	27.84%	12.44%
1:5	69.45%	21.49%	20.28%	45.71%	31.12%	12.59%
1:6	77.79%	33.28%	21.57%	50.71%	38.74%	14.75%
1:7	77.49%	45.15%	25.87%	61.83%	45.18%	14.78%
1:8	80.57%	52.17%	34.65%	79.48%	49.17%	16.49%
1:9	100.00%	62.81%	34.62%	100.00%	61.67%	16.51%
1:10	71.08%	45.78%	30.79%	87.91%	47.12%	15.49%

**Table 4 ijms-25-03993-t004:** Antioxidant activity of nanoparticles.

AgNPs Concentration (µg/mL)	AgNPs % of Inhibition			BHT % of Inhibition
	Leaves	Stem	Fruits	
1	35.10 ± 0.50	30.92 ± 2.05	33.27 ± 3.52	37.03 ± 3.28
10	41.37 ± 1.00	30.84 ± 0.00	32.29 ± 1.16	34.62 ± 0.37
100	53.17 ± 1.14	38.23 ± 0.57	32.29 ± 1.16	40.48 ± 0.64
250	62.41 ± 1.47 **	45.62 ± 0.74	35.49 ± 1.41	45.78 ± 1.05
500	79.04 ± 4.74 **	49.64 ± 0.24	40.38 ± 1.23	53.17 ± 0.45

** *p* < 0.005.

## Data Availability

Data is contained within the article.

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
