# Peer review of "Nanoremediation and Antioxidant Potential of Biogenic Silver Nanoparticles Synthesized Using Leucena’s Leaves, Stem, and Fruits"

_ijms, 2024, doi:10.3390/ijms25073993_

Round 1
Reviewer 1 Report
Comments and Suggestions for Authors
Aqueous extracts from different parts of the Leucaena leucocephala plant were utilized to synthesize silver nanoparticles (AgNPs), enhancing the concentration of phenolic compounds. These extracts served as effective bio-based materials for green nanomaterial synthesis. The resulting spherical nanoparticles displayed a negative surface charge, low polydispersity index (<0.3), and nanoscale size, as observed via TEM. EDS confirmed the presence of silver atoms. All three types of biogenic AgNPs demonstrated remediation capabilities against methylene blue and tartrazine dyes, with those derived from leaf extracts showing superior efficiency.
I would recommend some related references to books the introduction part
1)https://link.springer.com/chapter/10.1007/978-3-030-47400-3_15
2)https://link.springer.com/chapter/10.1007/978-981-19-7834-0_1 Macroporous Cryogel-Based Systems for Water Treatment Applications and Safety: Nanocomposite-Based Cryogels and Bacteria-Based Bioreactors
3) Photocatalytic Functional Materials for Environmental Remediation book
|
Table 1. Phenolics content of extracts obtained in different tempertures. The content is different for leaves fruites, stems but the SPR peak in figure 2 is the same for leaves fruits, stems. theoretically it should not be like this. Moreover the baseline even identical
Lines 294-300 4.5. AgNPs’ synthesis It is not clear to what extend the extract was duluted by water. please update the method description to make it reproducible.
Lines 123-130 Transmission electron microscopy (TEM) analysis revealed spherical AgNPs presenting an average diameter of 77,8 nm for the leaves (Figure 3a), 26,7 nm for the stem (Figure 1 3b) and 45,2 nm for the fruits (Figure 3c).... There is a contradiction in data obtaibed by TEM and UV-vis, as the SPR peak is the same the size distribution should be more or less the same.
|
Figure 3. TEM image of green AgNPs synthetized applying a. leaves; b. stem or c. fruits extract
TEM should be presented at the same magnification and moreover you must present at lower magnification that it were not or it were aggregated.
There references are not related to catalysis in discussion part and shoul be removed.
Lines 237-244 The fruits used in this studied included the seeds, as they were not removed. Extracts obtained from the seeds of Leucena have also already been reported as being able to allow the biogenic synthesis of AgNPs; these particles presented antimicrobial and antioxidant activities and selective optical sensing towards Fe3+ [36]. Polyssacarides from these seeds allowed the production of AgNPs capable of contributing to milk preservation; the nano- material exhibited antifungal and anticancer activities [42]. AgNPs produced using as green raw material Leucena’s extracts also proved to be useful to be applied in sensors to detect inorganic pollutants such as ions of the heavy metal mercury [43]
Author Response
Dear Reviewer 1,
Greetings!
First of all, we would like to thank you for your comments that contributed to improving the manuscript quality.
As requested, references (Pandikumar and Jothivenkatachalam, 2019; Sardar et al., 2020 and Savina et al., 2023) were added to the introduction (written in red). We have also added more references exemplifying that green AgNPs have already been proved as able to promote nanoremediation through catalysis (photocatalysis or chemical degradation) and adsorption.
Regarding remediation assay, references were provided, as requested. When it comes to the use of AgNPs at 1000 ppm as the start concentration, it was based on the work of Foster et al, and it is now mentioned in the manuscript (https://www.hindawi.com/journals/jnm/2019/9807605/).
Regarding Figure 2, we have committed a mistake and plotted the triplicate from one of the samples analyzed (leaves). Now the correct version of the Figure is displayed and in Discussion we address the difference of UV-Vis results associated with nanoparticles' size. These adjustments also address the aspects highlighted associated with Lines 123-130 “There is a contradiction in data obtained by TEM and UV-vis, as the SPR peak is the same the size distribution should be more or less the same”.
When it comes to Lines 294-300, the text was adjusted to state clearly to what extent the extract was diluted by water. The proportions presented refer to v/v and this information is now present in the manuscript.
Regarding lines 316-320 (AgNPs’ characterization): the text was adjusted, as requested, to present dilution performed before analysis.
When it comes to lines 245-256, the nanoparticles obtained using the extract from Leucena’s leaves exhibited antioxidant potential. So, they have the potential to be used not only in remediation, but also as antioxidant structures in biomedical field, for example. However, prior to that it is necessary to perform cytotoxicity assay and other experiments. The text “Regarding the antioxidant activity, it is of special interest to fight free radicals: the ones produced as a consequence of the metabolism and also the ones that are consequence of exposure to environmental contamination, immune response, ionizing radiation and so on [44]. These structures are associated to a large array of diseases [45] such as Parkinson’s and Alzheimer's disorders [46], asthma [47], atherosclerosis [48] and cancer [49], due to the fact that they, as reactive substances, can damage cell’s structure and impair biomolecule’s regular functioning. Antioxidants are used to mitigate free radicals’ production and action contributing to longevity in a healthy manner [45]. Nanoparticles can act as nano antioxidants offering advantages such as the possibility to be designed to perform targeted delivery and controlled release and also an increased bioavailability [45], which is especially useful in the medical field. They can favor, for example, cancer treatment [50] and wound healing [51]” is to give this idea. The text was adjusted to state it more clearly.
Regarding the action mechanism, it is now discussed. However, the data regarding this aspect is still preliminary. Some more experiments are necessary and we are already dedicating attention to that. We have mentioned this fact and added it as a future perspective. We have also mentioned the aspects regarding how we suggest to treat water after the remediation process is completed. As suggested, we have added the mentioned reference: https://www.mdpi.com/2227-9717/11/6/1661
Regarding Figure 3 the images were substituted to display a different magnification showing size difference among nanoparticles and how distant they were from the other ones. The manuscript also displays in this new version the size distribution graph.
When it comes to the references required to be removed from discussion (Lines 237-244), they were moved to introduction. They are important to explain to readers that there are other AgNPs made from Leucena’s parts to perform different tasks such as act as antimicrobial substances or constitute sensors to detect pollutants.
All figures were updated to improve quality allowing the readers to read all the information presented in them in a better manner. Other alterations were also performed to adjust the text according to the other reviewers’ suggestions.
Once more we would like to thank you for all the suggestions and comments.
Reviewer 2 Report
Comments and Suggestions for Authors
The article describes the synthesis of nanoparticles using an aqueous extract of leaves, stems and fruits of Leucaena leucocephala, which were used in further experiments to eliminate tartrazine contamination in 140 aqueous samples.
The article is written in clear language, the authors used adequate methods.
However, there are several comments about the work:
Figure 2 needs to be improved. The font in the axis labels needs to be corrected. It is necessary to write the units of measurement along the Y axis. Usually they write relative units. In the caption to the figure you should write something like “UV-vis spectroscopy AgNPs synthesized from Leucena’s leaves, stem and fruits extracts”
The authors further write: “Transmission electron microscopy (TEM) analysis revealed spherical AgNPs presenting an average diameter of 77.8 nm for the leaves (Figure 3a), 26.7 nm for the stem (Figure 3b) and 45.2 nm for the fruits (Figure 3c)". The question immediately arises: was only one microphotograph of each sample taken? Or several microphotographs? Then it is necessary to give the average value of nanoparticles and indicate the spread of values. If only one microphotograph was taken (which is strange), then where did such accuracy in measuring the diameter come from? After all, the photographs clearly show that the particles do not have a perfectly spherical shape.
Figure 4 requires improvement in image quality. The font is almost impossible to read, it is blurry. In figure 4b, the authors forgot to label the X-axis.
The authors write: “Scattering dynamic light (SDL) analysis revealed that the hydrodynamic diameter of green AgNPs synthesized from leaves was 181.57 nm. " It is necessary to indicate the statistical spread of nanoparticle size values. It is impossible for particles to be the same size and with such precision.
In figures 5 and 6, you need to write the units of measurement along the Y axis.
Author Response
Dear reviewer 2,
Greetings!
First of all, we would like to thank you for your comments that contributed to improving the quality of the manuscript.
Figure 2 was not displayed in its correct version. It was showing the triplicate from one of the samples analyzed (AgNPs from leaves). For that reason, it was corrected and substituted. This new version brings (a.u.) as the unit of measurement along the Y axis (a.u. = absorbance units). The caption of the figure was adjusted as required.
Regarding TEM, the samples were photographed during 1 hour each; so, there is more than one microphotograph available for each sample. The images were substituted to display a different magnification showing size difference among nanoparticles and how distant they were from the other ones. The manuscript also displays, in this new version, the size distribution graph. When it comes to the shape, it was updated considering the fact that different ones were present but most of the nanomaterial exhibited a quasi-spherical one; in discussion this aspect was also addressed.
Figure 4 was improved in quality and label of X-axis in Figure 4b, added.
Regarding Scattering dynamic light (SDL) analysis, the standard deviation for hydrodynamic diameter was added as requested. Zeta potential deviation is now presented too.
In figures related to nanoremediation assay, the unit of measurement along the Y axis was added (a.u.).
All figures were updated to improve quality allowing the readers to read all the information presented in them in a better manner. Other alterations were also performed to adjust the text according to the other reviewers’ suggestions.
Once more we would like to thank you for all the suggestions and comments.
Reviewer 3 Report
Comments and Suggestions for Authors
The manuscript presents a study on the biosynthesis of silver nanoparticles (AgNPs) using aqueous extracts from different parts of Leucaena leucocephala. The synthesized AgNPs are then evaluated for their efficacy in remediating methylene blue and tartrazine pollution in water, as well as their antioxidant activity. However, the quality of the manuscript is not suitable for publication in JMS. The results are listed after the other without sufficient discussion. This would be rather the style of a lab report than that of a scientific piece of work.
In my opinion, this manuscript is unsuitable for publication in the International Journal of Molecular Sciences.
Specific Comments:
1- The manuscript lacks a coherent structure and sufficient discussion of the results. Rather than scientifically presenting the results, they are listed without adequate interpretation. A more structured approach to presenting the results and discussing their implications in the context of existing literature is needed.
2- The novelty of the study is unclear, as there are numerous reported studies on the biosynthesis of AgNPs and their applications in pollutant remediation and antioxidant activity. The authors should clearly articulate the unique contribution of their work compared to existing literature.
3- The study mentions the synthesis of AgNPs using extracts from various parts of Leucaena leucocephala, but fails to discuss the potential differences in activity among these nanoparticles. A detailed analysis of the observed variations in activity would enhance the scientific significance of the study.
4- The figures presented in the manuscript ar poorly presented and require improvement.
5- XPS analysis should be provided for the synthesized AgNPs using different extracts and the size 5- distribution of the nanoparticles using TEM images should be presented.
6- The activity of the synthesized AgNPs should be compared with that of nanoparticles obtained through different chemical and biological methods reported in the literature. This comparative analysis would provide valuable insights into the effectiveness of the proposed method.
7- The language needs to be significantly improved considering the vocabulary and grammar.
Comments on the Quality of English Language
The language needs to be significantly improved considering the vocabulary and grammar.
Author Response
Dear Reviewer 3,
Greetings!
First of all, we would like to thank you for your comments that contributed to improving the quality of the manuscript.
1-The discussion of the results was extended to better address the data obtained and relate it to the one available in literature, providing a clearer interpretation.
2- The novelty of the study is now presented in the manuscript. The optimization in phenolics’ content through the described protocol allowed the production of AgNPs capable of remediating pollution caused by methylene blue and tartrazine. The AgNPs obtained using the extract from the leaves of Leucena exhibited better results for qe associated with dyes' adsorption performing the remediation in a manner independent of light direct incidence and reducing agents presence. It is an advantage when compared to the performance of other green AgNPs previously reported and a relevant feature aiming for large scale application to remediate industrial wastewater, for example. These nanoparticles also presented potential to be further investigated aiming application in the medical field once they exhibited antioxidant activity that surpassed the one exhibited by AgNPs from Leucena previously reported and the one from BHT. We have also added future perspectives to the manuscript.
3- The study now addresses the potential differences in activity among the synthesized nanoparticles aiming to enhance its scientific significance. References are provided to support the information presented in discussion.
4- All figures were substituted by new versions to improving quality.
5- XPS analysis is now present in the manuscript as requested, and size distribution of the nanoparticles using TEM images is displayed in this new version of Figure 3. We have also performed XRD and this new version also presents the results.
6- The activities (nanoremediation and antioxidant) are now compared with that of nanoparticles obtained through other green protocols to produce AgNPs reported in the literature.
7- The English language received attention to improve aspects related to grammar, spelling and to remove typos. The manuscript was gently revised by a native speaker: Prof. Adam Nakrewicz.
Other alterations were also performed to adjust the text according to the other reviewers’ suggestions.
Once more we would like to thank you for all the suggestions and comments.
Round 2
Reviewer 1 Report
Comments and Suggestions for Authors
Thank you for providing the requested information. all comments were addressed.
Reviewer 2 Report
Comments and Suggestions for Authors
The authors have made corrections to the manuscript; I believe that in this form it can be accepted for publication.